# A Novel RC-Based Architecture for Cell Equalization in Electric Vehicles

**Alfredo Alvarez-Diazcomas** [1,†], **Adyr A. Estévez-Bén** [2,†], **Juvenal Rodríguez-Reséndiz** [1,*,†], **Miguel-Angel Martínez-Prado** [1,†] **and and Jorge D. Mendiola-Santíbañez** [1,†]

1   Facultad de Ingeniería, Universidad Autónoma de Querétaro, Las Campanas, Querétaro 76010, Mexico; aalvarez78@alumnos.uaq.mx (A.A.-D.); miguel.prado@uaq.mx (M.-A.M.-P.); mendijor@uaq.mx (J.D.M.-S.)
2   Facultad de Química, Universidad Autónoma de Querétaro, Las Campanas, Querétaro 76010, Mexico; aestevez05@alumnos.uaq.mx
*   Correspondence: juvenal@uaq.edu.mx; Tel.: +52-442-192-1200
†   These authors contributed equally to this work.

**Abstract:** Nowadays, research on electric vehicles is increasing because they have the potential to decrease greenhouse-gas emissions dramatically in the transport sector. For these types of vehicles, the battery is one of the main components. The traction system needs a cell series connection to fulfill the energy requirements. Nevertheless, batteries differ from each other due to a normal dispersion in their capacity, internal resistance, and self-discharge rate. This paper presents a novel battery equalizer circuit using an RC-based topology to equalize two adjacent cells of a battery pack. It has the advantage of merging a resistor-based equalizer, a capacitor-based equalizer, and an RC-based equalizer in one circuit. In this way, it is possible to limit the current stress in the components of the circuit. The proposed method increases the equalization time by 35% for a threshold current of 4 A. However, it is possible design the system for another threshold current. Finally, the complexity of the controller is not compromised in the proposed architecture. The operation, analysis, and design of the architecture are presented and compared to the classic schemes. The theoretical analysis is validated through simulation results.

**Keywords:** electric vehicles; cell equalization; capacitor-based equalizers; battery management system

## 1. Introduction

There is currently a growing interest in Electric Vehicles (EVs). They have the most efficient engines when compared with Internal Combustion Engine (ICE) vehicles. Moreover, during its operation, greenhouse gases are not released into the atmosphere. Therefore, they represent a transportation possibility with a low-environmental impact [1].

The creation of EVs, ICE vehicles, and steam vehicles dates back to the late 19th century, and they fought fiercely for supremacy in the market. However, ICE vehicles were leading the market in 1920. At the end of the 20th century, electric vehicles resumed due to the environmental impact [2]. In recent years, interest has increased due to technological advances; for example, lithium-ion batteries have increased their capacity, lifetime, and safety with low costs [3].

Nevertheless, EVs still present many challenges to solve their extensive use, i.e., increasing the range, decreasing the charging time, and improving its transient performance to attract buyers. All these elements are closely related to the battery. The battery represents the most expensive, heavy, and bulky component in an EV. Therefore, its protection and proper use are significant [4].

The equalization of the cells is the biggest challenge to solve for the electronics industry [5]. In an EV, cells are stacked in a series, forming a battery bank to fulfill the requirements of the



propulsion system. However, the cells differ from each other due to a normal dispersion in their capacity, internal resistance, and self-discharge rate during manufacturing. Hence, the use of Battery Equalizer Circuits (BECs) is needed [6,7].

BECs take active measures to keep all the cells of the bank within an allowable range of voltage or State of Charge (SoC). Figure 1 illustrates the main classifications for the equalizers presented in the literature. This classification corresponds to fixed battery architectures. In recent years, a hot topic of research is the reconfigurable battery or software-defined battery. In this concept, the interconnections of the cells in a battery pack are reconfigured using switches. In this way, a great flexibility is achieved in the battery; its parameters such as the available power, operating voltage, etc., can be modified in real time. This flexibility is used to accommodate the battery and load requirements. Nevertheless, since EVs require a large-scale battery pack, this technology presents great challenges. The weight associated with the switches and drivers can reach 8% of the total system. Moreover, the switches are in the path of the power current. Therefore, the efficiency of the system is affected and thermal management is needed. Furthermore, the operation in real time is complicated since the current flowing through the switches is large. Finally, real-time control represents a big challenge since it presents a large number of switches—and therefore, a large number of possibilities to decide the battery configuration [8].

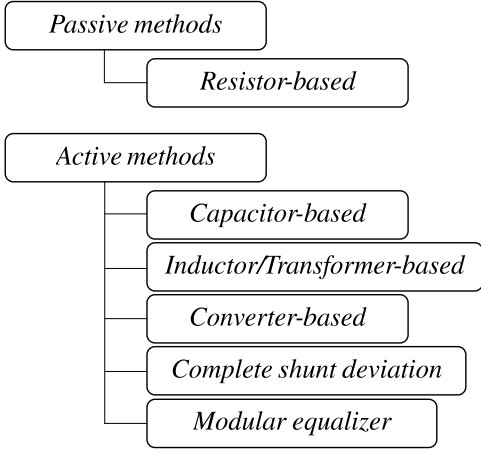

**Figure 1.** Classification of the Battery Equalizer Circuits (BECs).

This figure shows two strategies, the passive and active circuits. The passive circuits burn the excess energy in one cell through a resistor. Moreover, the active circuits transfer the excess energy between cells. Therefore, the active architectures are more appropriate for high-power applications such as EVs [6,7,9]. There are mainly three solutions for passive equalizers. One method consists of a fixed shunting variable resistor. The current flow through the resistor is controlled, acting on its value [7]. Moreover, reference [10] presents the switched resistor topology. In this scheme, the cell is in shunt with a resistor and a switch. The switch is activated to let the current flow through the resistor when the battery reaches the full-charge voltage. Furthermore, reference [11] proposes an analog shunting scheme. This is the most efficient within the passive equalizers since a transistor to bypass the current is used. The limitation of this architecture is that it requires drivers with analog capabilities. Passive techniques are cheap and straightforward but present low efficiency. Hence, active equalizers were developed to overcome the low-efficiency limitation. The capacitor-based architectures represent an attractive solution since they are cheap, small in size and volume, and do not require a complex controller [6,9]. Reference [12] proposes the switched capacitor. In this scheme, $n - 1$ capacitors and $n$ Simple Pole Double Throw (SPDT) switches are needed to equalize $n$ cells. The equalization occurs naturally by applying a Pulse-Width Modulation (PWM) signal to the switches. In reference [13], the single-capacitor topology decreasing the component count is presented. In this architecture, the equalization time is increased, since one capacitor handles all the equalization. Reference [14]

presents the double-tiered switched capacitor where two layers of capacitors are connected in parallel with the battery pack. In this structure, the equalization time is decreased by 25% when compared to the switched capacitor. Despite its advantages, all these architectures present high-current stress in the switches [10,12]. In this paper, a novel topology of the RC-based equalizer is proposed to overcome the surge current in the cells without compromising the complexity of the controller. In this way, the cells are protected, and the health of the system is improved. These highlights are achieved by combining the two classic equalizer schemes.

This topology is especially suitable for battery banks with a high-dispersion voltage between cells. As stated in [15], the operating voltage is affected by internal parameters such as capacity and internal resistance. Furthermore, temperature and charge/discharge current are external parameters that affect the voltage as well. Therefore, this equalization variable does not reflect the internal state of the cell accurately [16]. Moreover, the voltage is more affected if aged cells are used in the battery pack. For these reasons, there can be large transient voltages in the cells. Therefore, a high-dispersion of voltage across the battery pack will occur [9,17]. The large voltage difference induced can damage the cell if classic capacitor-based topologies are used due to the high-equalization current demanded. Furthermore, when modular schemes are used, there are equalizers between modules where conditions are more prone to large voltage differences since they group several cells.

In the literature, transient current peaks have been limited with snubber circuits [18,19]. This solution is quite cheap and straightforward. However, the sole purpose of this approach is to limit the current. This paper presents a topology that can emulate the operating principle of a switched resistor-based equalizer and a switched capacitor-based equalizer, and can limit the current if the safety of the battery is compromised. A high-level controller can use this flexibility to perform the strengths of the system in every condition. In Section 2, the proposed architecture is presented, and its modes of operation are explained; in Section 3, the steps to design the converter are illustrated; in Section 4, the proposed converter is compared to the classic topologies of capacitor-based and resistor-based BECs. In Section 5, the simulation results are shown. Finally, conclusions are presented and the main generalizations of the paper are given.

## 2. Proposed RC-Based Equalizer

Figure 2 presents the classic resistor and capacitor-based BECs. Both architectures work, forcing the passive element in parallel with the voltage source to achieve equalization. The controller regulates the state of the switches to achieve equalization. The capacitor-based scheme does not require a complex controller since a PWM signal applied in the switches is enough. These signals force the capacitor to be in a shunt with one cell or its adjacent. The energy is transferred between cells naturally.

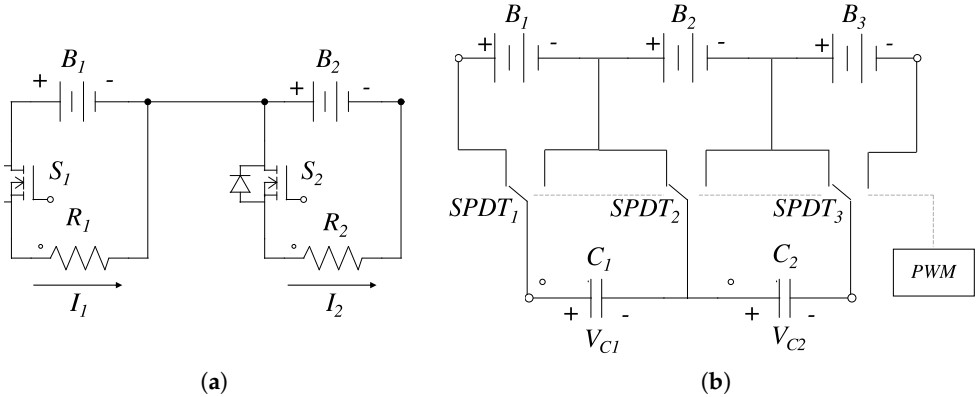

(**a**)　　　　　　　　　　　　　　　　　　　　　　　(**b**)

**Figure 2.** Classic schemes of BECs: (**a**) resistor-based equalizer; (**b**) capacitor-based equalizer.

The proposed topology is illustrated in Figure 3. It is very similar to the classic capacitor-based topology. Nevertheless, a resistor in series with the capacitor and two switches in parallel with both

elements, respectively, are added. In this way, a very flexible scheme that can operate as a resistor-based equalizer, a capacitor-based equalizer, or an RC-based equalizer is obtained. The last topology has proved to be very helpful in conditions of a significant voltage difference between adjacent cells. Next, these three modes of operation are discussed.

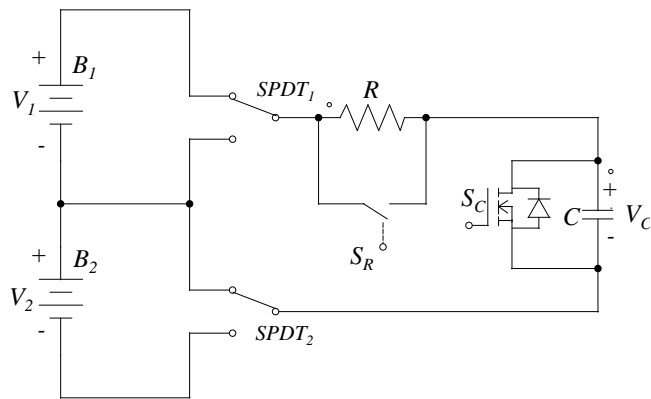

**Figure 3.** The proposed RC-based battery equalizer.

### 2.1. Proposed RC-Based Equalizer Operating as a Resistor-Based Equalizer

Figure 4 illustrates the proposed topology operating as a resistor-based equalizer. The switch $S_R$ is off, while $S_C$ is on. In this way, the cell is in parallel with the resistor. The SPDT switches serve to establish the proper cell. In this method, the voltage of the cells is maintained inside an allowable range. The resistor dissipates the excess energy in the form of heat. This strategy is straightforward and cheap. However, it presents low efficiency, poor equalization time, and requires a heat management system. Therefore, it it is not appropriate to use in EVs [20,21].

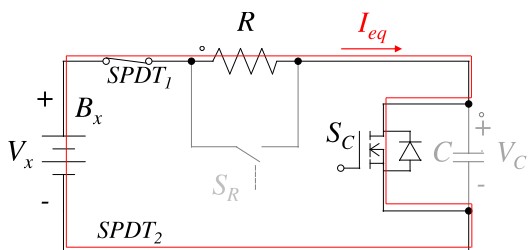

**Figure 4.** Resistor-based mode of operation.

### 2.2. Proposed RC-Based Equalizer Operating as a Capacitor-Based Equalizer

The capacitor-based architecture overcomes the limitations present in the resistor-based scheme. The proposed topology can operate as a capacitor-based equalizer, as shown in Figure 5. In this strategy, the switch $S_R$ is on, and $S_C$ is off. In this way, the resistor is short-circuited, so it does not influence the operation of the circuit. By applying a PWM signal in the SPDT switches, the behavior of the classic capacitor-based topology is achieved. Capacitors are very important in this scheme, as its name suggests, since one of these passive elements is used to transfer energy between adjacent cells. Figure 6 shows the steady-state waveforms of the voltage in the capacitor $V_C$ and the equalization currents $I_{eq\_1}$ and $I_{eq\_2}$ during the operation of the converter. In this waveform, it is assumed that the operating voltage in cell $V_1$ is greater than $V_2$; on the contrary, the equalization currents are negative. This strategy is simple to understand and control since only a PWM signal in the SPDT switches is enough for two adjacent cells to reach the equalization in a natural manner [13,22,23]. Nevertheless, in this method, large current stresses in the cell and a large equalization time are present [24,25].

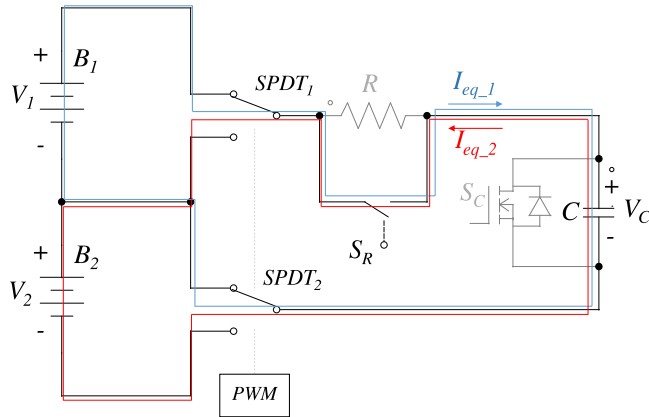

**Figure 5.** Capacitor-based mode of operation.

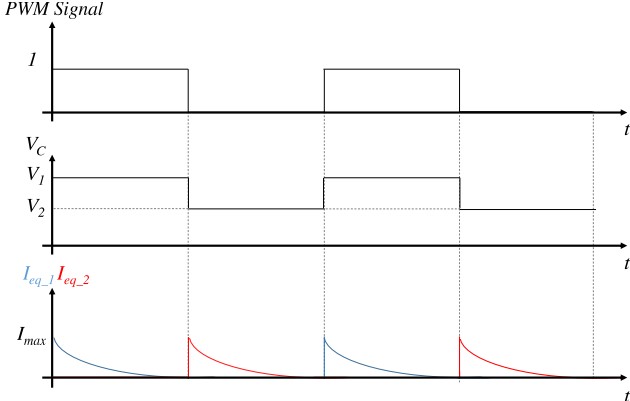

**Figure 6.** Steady-state waveforms of classic capacitor-based mode of operation.

### 2.3. Proposed RC-Based Equalizer Operating as an RC Circuit

Figure 7 depicts the proposed topology operating as an RC circuit. This is the last possible mode of operation of the circuit. The opportunities present in this architecture are very interesting. First, in this mode of operation, both switches $S_R$ and $S_C$ are off all the time. Therefore, when the PWM signal is applied, each cell is forced in parallel with an RC network. This is a widely studied circuit, and it is very helpful to overcome the limitations present in the capacitor-based architecture. In general, the waveforms in this mode of operation are the same as the ones presented in Figure 6, but with another maximum current, as discussed next.

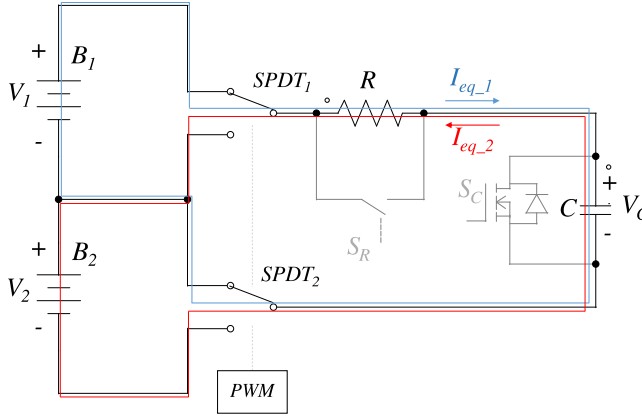

**Figure 7.** RC-based mode of operation.

In the above figure, $v_C(t)$ is the voltage in the capacitor, $V_2$ is the voltage in the cell $B_2$, $V_1$ is the voltage in the cell $B_1$, $R$ is the resistor, $C$ is the capacitor, and $R_{eq}$ includes all the equivalent resistances of the battery, capacitor, path, and switches.

The circuit is analyzed as an RC network, which is in steady state due to the action of one voltage source, and at a certain time the voltage source is changed. This is a classic circuit to study in a circuit engineering course. Therefore, it is easy to find Equation (1), which describes the behavior of the voltage in this circuit [26,27]. Moreover, Equation (2) represents the behavior of the current in a capacitor. Therefore, Equation (1) is substituted in Equation (2) giving Equation (3), which describes the current of the circuit.

$$v_C(t) = V_2 + (V_1 - V_2)e^{-t/(R+R_{eq})C} \tag{1}$$

$$i_C(t) = C\frac{dv_c(t)}{dt} \tag{2}$$

$$i_C(t) = \frac{V_1 - V_2}{R + R_{eq}}e^{-t/(R+R_{eq})C} \tag{3}$$

The surge current depends on the value of the resistance and voltage difference between the cells. The initial current is greater with a lower resistance, while the voltage difference is directly proportional to the surge current. Therefore, the use of a fixed capacitor limits the system for its low intrinsic resistance and its compromise with the surge current. Precisely, this circuit has the possibility to limit the surge current when a large voltage difference exists between adjacent cells by acting as an RC network. On the contrary, when a low voltage difference between cells exists, a low resistance is desirable to shorten the equalization time. Therefore, the flexibility of the topology helps to overcome the limitations of the classic capacitor-based equalizer.

In this paper, the operation of the equalizer is proposed, as illustrated in Figure 8. When the voltage difference between adjacent cells is greater than a threshold (given by the application and the batteries used), the RC-based mode is used to limit the surge current and therefore protect the cells. Nevertheless, when the voltage difference falls below the threshold, it is possible to use the capacitor-based mode to increase the equalization current and therefore decrease the equalization time. In this way, the flexibility of the circuit is used to overcome the limitations of the classic capacitor-based equalizer.

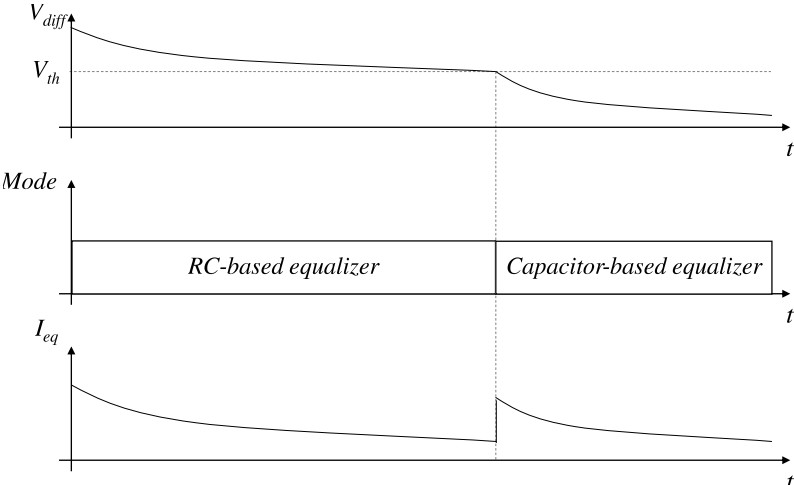

**Figure 8.** Functional diagram of operation for the proposed topology.

## 3. Design of the Proposed Equalizer

This section aims to provide a methodology for the proper design of the proposed equalizer. It is important to note that the design process must solve the value of the resistance, the value of the capacitor, the switching frequency of the SPDTs switches, and the threshold voltage to activate the

$S_R$ switch. First, it is important to highlight that the methodology proposed here takes into account an 18650 battery cell. These cells present a characteristic curve as depicted in Figure 9 [28]. At 70% of the capacity of the battery, the voltage behaves very smoothly. However, there are exponential regions where a small change in capacity induces a large change in voltage. Therefore, in those regions, it is difficult to operate the classic capacitor-based equalizer due to the high current stresses in the cell [29,30].

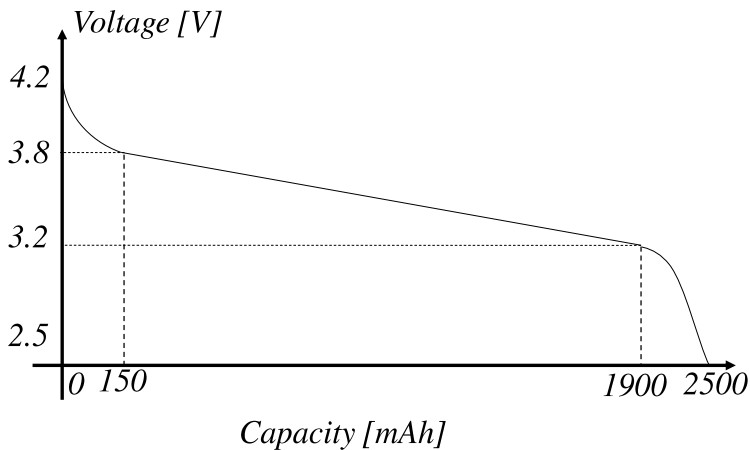

**Figure 9.** Characteristic curve of the 18650 battery.

The first element to consider is the resistance. If a resistor-based equalizer wants to be emulated, it is important to burn the excess energy given to the cell through a resistor. Therefore, it is important to have the capacity to bypass, at least, all the charge current. The data sheet of the 18650 battery provides the nominal and fast charge current. For this analysis, a nominal charge current of 1.25 A and a 4.2 V end-of-charge voltage are considered. These values lead to a resistance of 3.36 Ω to bypass all the charge current. However, in practical situations the current, voltage and resistance vary of its nominal value. Therefore, a 3 Ω value of resistance is selected with the possibility to dissipate at least 5.88 W. The resistor burns the excess of energy until the operating voltage falls below the full-charge voltage.

The switching frequency is a crucial parameter to select in a power converter. It affects nearly every performance characteristic of the converter. At higher frequency, converters presents a poor efficiency and current limit accuracy is negatively affected and increases the cost of the output filter. Nevertheless, it induces benefits in the size and the transient response of the converter [31,32]. Therefore, it is important to find a middle ground for this parameter. An acceptable switching frequency is around 50 kHz and is currently an achievable value for most switches.

The capacitor value $C$ depends on the selected switching frequency. The value of the capacitor decreases the switching losses in the circuit if selected properly. For this purpose, the current in the switching transition must be zero. Based on an analysis of Equation (3), the time constant ($\tau$) is $(R + R_{eq})C$, and it is known that the value of current is zero at $5\tau$. Therefore, the commutation of the switches after $5\tau$ is needed to neglect the switching losses. In this case, the signal presents a period of 20 µs, and $R_{eq}$ is estimated to be 110 mΩ. Based on the solution of Inequation (4), the capacitor C must be less than 1.153 µF. The considered value of the capacitor is 1 µF for convenience with commercial standards.

$$20\mu > 5(R + R_{eq})C. \tag{4}$$

Finally, the threshold voltage ($V_{th}$) to activate the switch $S_R$ is obtained. It is important to analyze the Figure 8 and Equation (2) to notice the impact of the threshold voltage. When the voltage difference falls below the threshold voltage, the mode that emulates the classic capacitor-based equalizer is activated with an initial current surge of $V_{th}/R_{eq}$. Therefore, the threshold voltage is selected in such a way that in this transition it does not induce a dangerous current in the cells.

The experience of the designer in the application is needed to guarantee proper operation and the protection of the cells. First, it is important to guarantee that the current of the cell is not exceeded. Based on the data sheet of the battery, the fast charge current is 4 A. Taking into account that the estimated $R_{eq}$ was 110 mΩ, the maximum threshold voltage allowed is 0.44 V. In order to have a margin of safety, a threshold voltage of 0.4 V is selected. Figure 10 shows a flowchart of the design process to illustrate the steps in a clearer way.

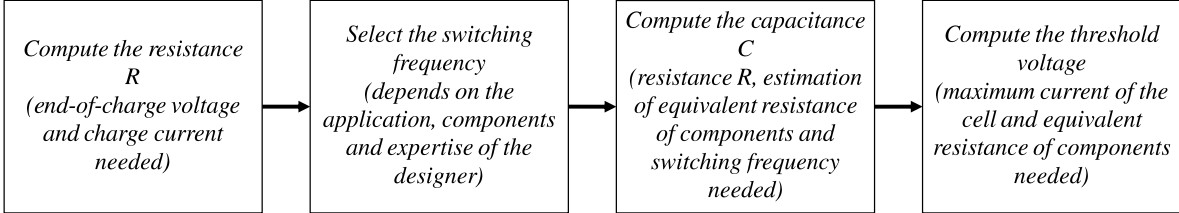

**Figure 10.** Flowchart of the design process.

The control system depends on the mode of operation established. The signal of the operating mode can be manually changed by the user or can be modified by a high-level controller that decides the best mode given certain conditions. If the resistor-based mode of operation is used, the switches must behave according to the truth table appreciated in Table 1. Signals $S_1$ and $S_2$ come from the comparison of the operating voltage of both cells with the end-of-charge voltage. They are 1 if the voltage is above the end of the charge voltage. Moreover, the signal $S_{ch}$ is a signal that controls the disconnection with the charge source. The implementation of this truth table is depicted in Figure 11.

**Table 1.** Truth table of the switches in resistor-based mode of operation.

| $S_1$ | $S_2$ | $SPDT$ | $S_C$ | $S_R$ | $S_{ch}$ |
|---|---|---|---|---|---|
| 0 | 0 | x | 0 | 0 | 0 |
| 0 | 1 | 0 | 1 | 0 | 0 |
| 1 | 0 | 1 | 1 | 0 | 0 |
| 1 | 1 | x | 0 | 0 | 1 |

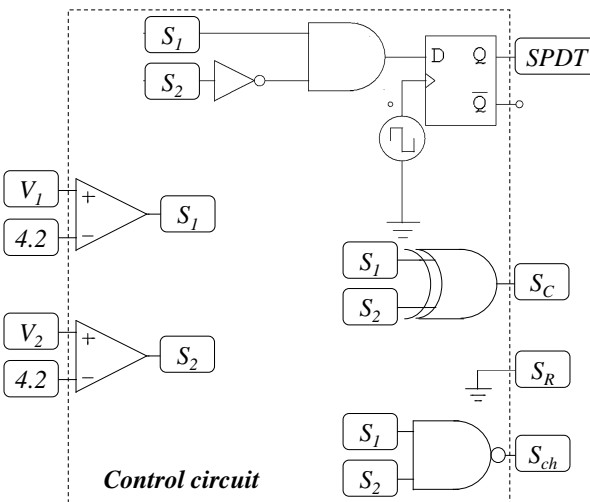

**Figure 11.** Control circuit for the resistor-based mode of operation.

Furthermore, Table 2 shows the truth table for the RC mode of operation. In this case, the signal $S$ is the difference in voltage between adjacent cells compared with the threshold voltage. This signal is true if the voltage difference is below the threshold voltage. The value of the SPDT switches follows

the PWM signal independently of the value of *S*. Moreover, the charging source is disconnected when both cells reach the end of charge voltage, as in the previous case. The implementation of this truth table is shown in Figure 12.

**Table 2.** Truth table of the switches in resistor-based mode of operation.

| *S* | *SPDT* | $S_C$ | $S_R$ |
|-----|--------|-------|-------|
| 0 | PWM | 0 | 0 |
| 1 | PWM | 0 | 1 |

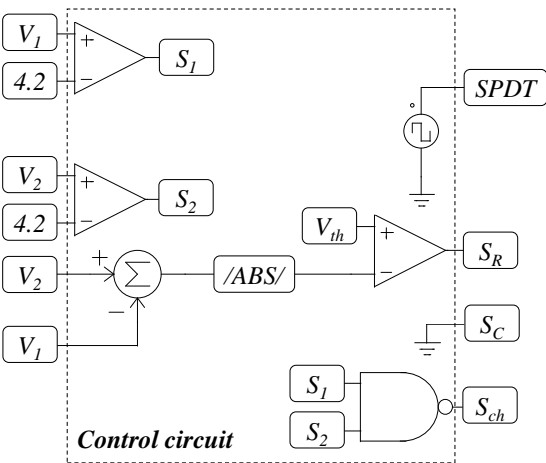

**Figure 12.** Control circuit for the RC-based mode of operation.

## 4. Comparison

This section helps to understand the benefits and limitations of the proposed equalizer with the other equalizers present in the literature. The main parameters to evaluate in an equalizer are its efficiency, equalization time, element count, and stress in components. First, a power loss analysis is performed. In power converters, there are two main components of the power losses, the conduction losses, and the switching losses. The conduction losses are due to the flow of the current through the series resistance of the real components. Equation (5) estimates the conduction losses that take into account the series resistance of every component present in the circuit.

$$P_{L\_Cond} = I^2(D(R_{SV_1} + R_{on\_SPDT_1}) + (1-D)(R_{SV_2} + R_{on\_SPDT_2}) + R_{DS(on)S_R} + R_{ESR(C)}), \quad (5)$$

where *I* is the current in the network, *D* is the duty cycle of the SPDT switches, $R_{SV_X}$ is the internal resistance of the correspondent battery cell, $R_{on\_SPDT_X}$ is the correspondent resistance when the SPDT switch is activated, $R_{DS(on)S_R}$ is the drain-source on-resistance of the MOSFET $S_R$, and $R_{ESR(C)}$ is the Equivalent Series Resistance (ESR) of the capacitor *C*. If $R_{SV_1}$ and $R_{on\_SPDT_1}$ are considered equal to the $R_{SV_2}$ and $R_{on\_SPDT_2}$, respectively, Equation (5) is reduced to Equation (6). This equation is used for the capacitor-based mode. For the RC-based mode, the term $R_{DS(on)S_R}$ for *R* is substituted since the switch $S_R$ is off and the current is flowing through the resistor. This term is completely suppressed in a classic capacitor-based topology; hence, this scheme presents lower conduction losses than the proposed topology. In the resistor-based mode, the objective is to burn the excess energy; therefore, there is no point in performing an efficiency analysis of this scheme.

$$P_{L\_Cond} = I^2(R_{SV_1} + R_{on\_SPDT_1} + R_{DS(on)S_R} + R_{ESR(C)}). \quad (6)$$

On the other hand, Equation (7) represents the switching power losses. $V_X$ is the voltage of the cell, 1 or 2 as appropriate; *I* is the current in the network; $f_{sw}$ is the switching frequency; $t_r$ is the rise

time of the switch; and $t_f$ is the fall time of the switch. Since the converter was designed to apply the switching frequency at zero current, as discussed in the previous section, it is demonstrated that the switching losses in the proposed topology are negligible.

$$P_{L\_Sw} = V_X I f_{sw}(t_r + t_f). \tag{7}$$

Equation (8) represents the losses in the resistor-based equalizer. $I$ is the charge current and $R$ is the resistor where the excess energy is burned. A low efficiency is presented when compared to the capacitor-based mode since $R$ is much greater than the term $R_{SV_1} + R_{on\_SPDT_1} + R_{DS(on)S_R} + R_{ESR(C)}$. Furthermore, Equation (9) illustrates the losses in the classic capacitor-based equalizer. It is assumed that the switching power losses are negligible. It can be noted that it is similar to the capacitor-based mode, but the term $R_{DS(on)S_R}$ is suppressed. Hence, when compared to the proposed topology, it is more efficient. However, the values of the resistances in the circuit are needed to accurately compute the percentage of improved efficiency.

$$P_{L\_Cond} = I^2 R, \tag{8}$$

$$P_{L\_Cond} = I^2 (R_{SV_1} + R_{on\_SPDT_1} + R_{ESR(C)}). \tag{9}$$

A typical configuration is analyzed to put these equations in perspective. A common cell in this type of application is the 18650 battery, which has a maximum allowed current of 4 A. Therefore, a safe equalization current is 3.5 A. Furthermore, the conduction resistance of the elements $R_{SV_1}$, $R_{on\_SPDT_1}$, and $R_{ESR(C)}$ can be estimated at 110 mΩ; and $R$ is established, in the previous section, as 3 Ω. For these values, the power loss for the resistor-based equalizer is 36.75 W. In the case of the capacitor-based equalizer, a maximum instantaneous power loss of 1.3475 W is obtained. In this case, since the current is a decreasing exponential function, the power losses will also behave in this way.

On the other hand, in the proposed equalizer, there is a first stage where the converter operates as an RC circuit. If a voltage difference between cells of 0.5 V is present in the circuit, a maximum current peak of 0.16 A is demanded from the cells, as seen in Equation (3). In this way, the instantaneous power losses have a maximum peak of 2.84 mW. When the voltage difference between the cells falls below a threshold, which guarantees a safe induced current in the cells, then the capacitor-based mode of operation of the proposed equalizer is activated. In this mode, a power loss equal to that of the classical capacitor-based scheme is obtained. It can be concluded that the proposed equalizer also limits the power losses in the circuit. Hence, the proposed topology should be used when there is a high voltage dispersion between cells.

Once the efficiency analysis was performed, the proposed topology is compared with the existing equalizers. The resistor-based equalizers burn the excess energy of the cell with higher operating voltage through a resistor. For this reason, they are known as cell-to-heat equalizers. The energy burned is controlled by means of a switch. If it is needed to burn energy, the switch is on to allow the current to pass. The merits of this topology are its simplicity and low cost. However, they present a very poor efficiency and large equalization time [6,7].

On the other hand, the switched capacitor is an architecture of equalizer where these passive elements are used to equalize two adjacent cells. The capacitor is forced in parallel with the cell with higher voltage to extract some of the energy. When it is completely charged, it is forced in parallel with the cell with lower voltage. In this way, the energy flows from the capacitor to the cell. This operation is achieved by applying a PWM signal in the switches. Therefore, one of its main merits is the simplicity of its controller. Moreover, they present a good efficiency and are cheap. However, they present a large equalization time and a large number of switches and can present high stress in the components of the circuit [9,10,12].

Finally, the proposed topology combines the two abovementioned structures. In this scheme, it is possible to emulate the switched resistor-based and the switched capacitor-based equalizers, and it is possible to limit the stress current in the circuit. This behavior is achieved by adding a resistor to the current path when there is a large voltage difference between adjacent cells. Moreover, when a

threshold voltage is reached, the resistor is short-circuited to improve efficiency and decrease the equalization time. The merits and demerits of every scheme are summarized in Table 3.

**Table 3.** Comparative analysis of several BECs.

| Equalizer | Equalization Principle | Components for $n$ Cells | Power Losses | Remarks |
|---|---|---|---|---|
| Switched resistor [7] | The switch is activated when it is desirable to burn an excess of energy | $n$ resistors $n$ switches | $I^2R$ | Is cheap, small in size, and simple to control and design. However, high power losses and a large equalization time is presented, and a thermal management system is needed. |
| Switched capacitor [13,22] | A PWM signal is applied to the SPDT switches leading to equalization between two adjacent cells | $n$ SPDT switches $n-1$ capacitors | $I^2(R_{SV_1} + R_{on\_SPDT_1} + R_{ESR(C)})$ | Presents low power losses and a simple controller. However, a large number of switches and a large equalization time are required. |
| Proposed equalizer | A PWM signal is applied to the SPDT switches leading to equalization between cells. A resistor is included in the path if a threshold voltage is surpassed | $n$ SPDT switches $2n$ MOSFETs $n-1$ capacitors $n-1$ resistors | $I^2(R_{SV_1} + R_{on\_SPDT_1} + R_{DS(on)S_R} + R_{ESR(C)})$ | Is simple to control and can limit the current of the cells. However, a large number of switches and a large equalization time are present. |

## 5. Simulation Results

This section shows the simulation results that validate the proposed equalizer. The software used was PSIM (version 9.0.3, Powersim, Rockville, United States of America). This software is designed specifically for its use in power electronics and motor drive simulations. The values of the components are summarized in Table 4.

**Table 4.** Value of the components used in the simulation.

| Component | Value |
|---|---|
| Battery cell | $Q$ = 2500 mAh, $I_{ch}$ = 1.25 A, $I_{max}$ = 4 A |
| Resistor | 3 Ω |
| Capacitor | 1 μF |
| Threshold voltage | 0.4 V |
| Equivalent resistance | 110 mΩ |

The waveforms obtained in the resistor-based mode of operation are depicted in Figure 13 in a constant-current charge method. The current passes through both cells, while the operating voltage of both cells are below the full-charge voltage. When $B_1$ reach the full-charge voltage, the current is redirected to the resistor. Once the voltage $V_1$ falls the current is forced through the cell again. In this case, it is important to limit the switching frequency for a possible implementation. Finally, the charge current ($I_{ch}$) is disconnected when the cell $B_2$ reaches its full-charge voltage.

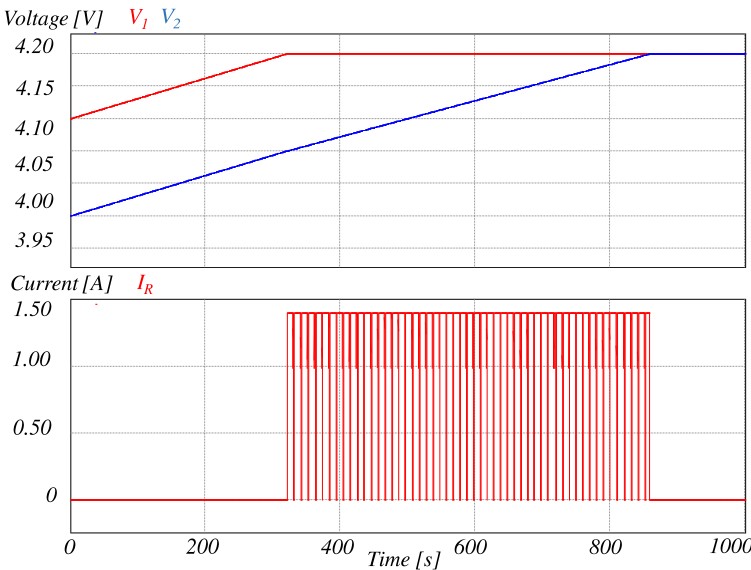

**Figure 13.** Resistor-based mode of operation with constant-current charge method ($I_{ch}$ = 1.25 A).

Another standard method of charge is the constant voltage. In this case, the voltage is fixed during the charge of the battery bank. Figure 14 shows the results obtained using the resistor-based mode and the constant-voltage charge method. When one cell exceeds the full-charge voltage, the same operating principle of the previous case is applied. This behavior is illustrated on the left side of the figure. On the right side the charge current is decreasing until it falls below a threshold value when the voltage charge ($V_{ch}$) is disconnected. It is concluded that the equalization in both methods of charge is achieved with the same principle. When one cell reaches the full-charge voltage, the excess energy is burned through the resistor to protect the cell against overcharge and to obtain the same operating voltage in both cells.

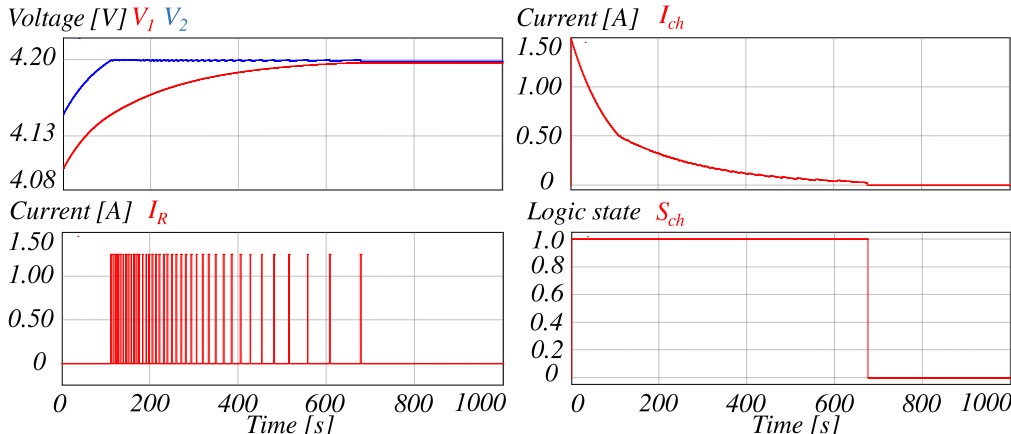

**Figure 14.** Resistor-based mode of operation with constant-voltage charge method ($V_{ch}$ = 8.4 V).

Figure 15 illustrates the RC-based mode of operation of the proposed equalizer. The initial condition presents both cells with a difference of voltage of 0.5 V in its lower limit of charge. Since this voltage difference is above the allowable threshold, the resistor is connected in the network to limit the current. In this way, the cells are protected against surge currents that surpass its safe limit current. Once the voltage difference falls below the established 0.4 V, the capacitor mode is activated. The safe operation of the equalizer for a wide span of voltage difference between cells is achieved with this operation. Moreover, Figure 16 shows a zoom in the equalization current. The theoretical analysis is

validated since the current is limited and the switching of the SPDT switches is done when the current is almost zero. This behavior decreases the switching losses significantly.

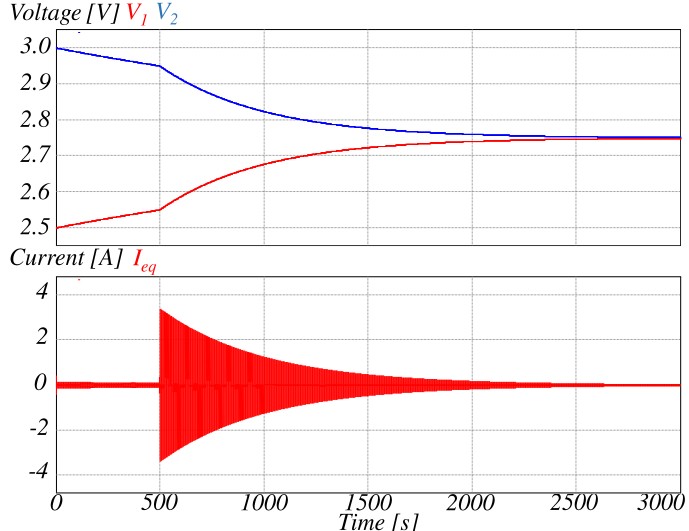

**Figure 15.** RC-based equalizer mode of operation.

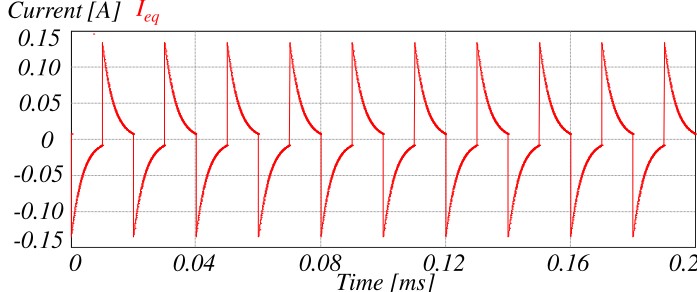

**Figure 16.** Zoom of the equalization current of the proposed converter operating in the RC-based mode.

Moreover, the simulations of the classic schemes of equalization are presented. Figure 17 illustrates the results obtained in a classic resistor-based equalizer. The charge current is bypassed through the resistor once the full-charge voltage is reached. This state is maintained until all cells reach the full-charge voltage. In this way, the cell is protected against overcharge during the charge process. It is noted that the excess energy is transformed in heat. Hence, a proper thermal management system is required.

Figure 18 shows the behavior of the classic capacitor-based equalizer in the same conditions under which the RC-based mode of the proposed topology was tested. A similar behavior is obtained, but it can be noted that the current in the beginning of the equalization process is not limited. A zoom of the equalization current is illustrated in Figure 19. The 4-A current limit of the cell is surpassed. This negatively impacts the aging process of the cells. Moreover, this method requires a lower equalization time when compared to the RC-based mode of operation of the proposed topology. It was obtained that the classic capacitor-based equalizer is 26% faster. However, this speed is achieved by extracting a current greater than the maximum current allowed for the cells.

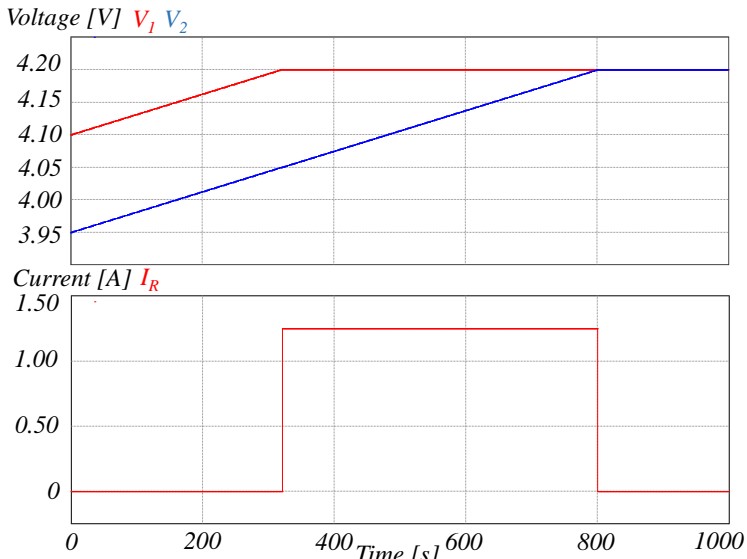

**Figure 17.** Classic resistor-based equalizer with constant-current charge method ($I_{ch}$ = 1.25 A).

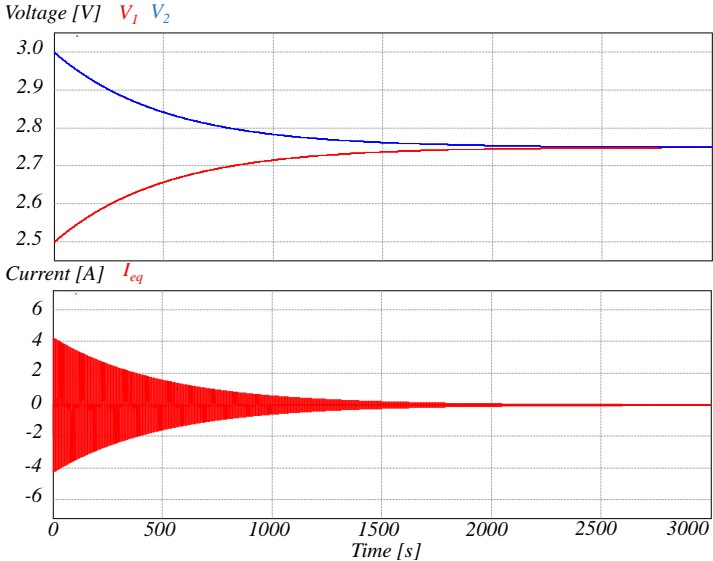

**Figure 18.** Classic capacitor-based equalizer.

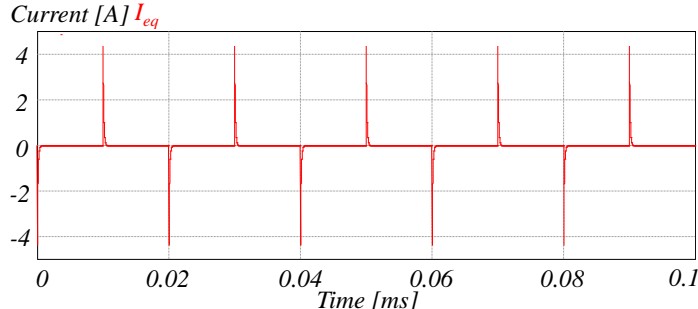

**Figure 19.** Zoom of the equalization current of the classic capacitor-based equalizer.

## 6. Conclusions

In this work, an RC equalizer for its application in EVs is proposed to achieve a homogeneous operating voltage in the cells of a battery bank. A very flexible topology that can be operated as

resistor-based, capacitor-based, or RC-based is achieved. Therefore, a specific mode of operation based on the merits of each equalizer in a specific scenario can be achieved. Moreover, the RC mode of operation proved very useful in limiting the current in the transfer of power between two adjacent cells. In this way, it overcomes the limitations of the classic capacitor-based equalizer and can be used in battery banks with a high dispersion of voltage.

This paper gives a detailed analysis of the operation and design of the converter for several modes of operation. A simple control strategy is presented to regulate the system. Therefore, it can be concluded that the simplicity of the controller is not compromised to overcome the limitations of the capacitor-based equalizer. The software PSIM was used for the validation of the proposed architecture and the theoretical analysis has been verified. The simulation results demonstrate the proper operation of the converter in the several modes designed. Moreover, how the current is limited in the case of a significant voltage difference between adjacent cells is illustrated. This converter has proved to be a low-cost solution for the equalization of cells within a battery bank in EVs, especially in battery banks with high-dispersion in voltage between cells.

**Author Contributions:** Conceptualization, A.A.-D. and J.R.-R.; methodology, J.R.-R. and M.-A.M.-P.; software, A.A.-D. and A.A.E.-B.; validation, A.A.-D., A.A.E.-B., J.D.M.-S., and M.-A.M.-P.; formal analysis, J.R.-R. and J.D.M.-S.; investigation, A.A.-D. and A.A.E.-B.; resources, J.R.-R., M.-A.M.-P., J.D.M.-S., and A.A.E.-B.; data curation, A.A.-D., J.R.-R., J.D.M.-S., and A.A.E.-B.; writing–original draft preparation, all authors; writing–review and editing, all authors. All authors have read and agreed to the published version of the manuscript.

**Funding:** This research was funded by CONACYT and PRODEP.

**Conflicts of Interest:** The authors declare that there is no conflict of interest.

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
