# Peer review of "A Novel RC-Based Architecture for Cell Equalization in Electric Vehicles"

_energies, doi:10.3390/en13092349_

Round 1
Reviewer 1 Report
The work is interesting. I have the following comments that need to be addressed for further consideration for publication.
The reviewer should be consistent with terms, such as “vehicles” and “cars”.
The reviewer strongly agrees that clean energy is needed, and ICE vehicles are replaced in this case. However, one important aspect to promote clean vehicles is because of the drop in cost and maturity of batteries. The authors are recommended to strengthen this argument in the introduction and include this reference to support this claim: Lai, C.S. and McCulloch, M.D., 2017. Levelized cost of electricity for solar photovoltaic and electrical energy storage. Applied Energy, 190, pp.191-203.
English is a major concern with several grammatical errors. The paper needs to be proofread by native English speakers or authors should seek for professional language editing service. Some errors include:
“In [9] is presented the switched capacitor” should be corrected to “Reference [9] presents….”
“In this scheme are necessary one capacitor and two switches to equalize two adjacent cells.”
“It is switched a capacitor between two adjacent cells”
The reviewer questions the novelty and meaningfulness of this work.
The claimed contribution of this work is: “A novel topology of RC-based equalizer is proposed to overcome the surge current in the cells without compromising the complexity of the controller. In this way, the cells are protected and it is improved the health of the system.
Generally, a snubber circuit can be used to suppress the inrush or surge current. Others have looked into ways for achieving this as follows:
Mizanur, R., Khan, S., Rahman, A., Hrairi, M., Ferdaus, M.M. and Shahid, Z., 2014, November. A Battery Charge Balancing system with reducing inrush high spike current for electric vehicle. In 2014 IEEE International Conference on Smart Instrumentation, Measurement and Applications (ICSIMA) (pp. 1-6). IEEE.
Kim, H.S., Park, K.B., Seong, H.W., Moon, G.W. and Youn, M.J., 2011, May. Flyback battery equalizer with voltage doubler rectifier. In 8th International Conference on Power Electronics-ECCE Asia (pp. 291-295). IEEE.
The authors need to strengthen the literature review and in particular, the contribution of the work.
Section 4 claims “to help to understand the benefits and limitations of the proposed equalizer with the other equalizers present in the literature.” Apart from the explanation, I suggest including some quantitative studies (e.g. total power loss) to support the proposed technique with comparison to the types of Equalizer identified in Table 3.
Author Response
Reviewer 1. The work is interesting. I have the following comments that need to be addressed for further consideration for publication.
- The reviewer should be consistent with terms, such as “vehicles” and “cars”.
We appreciate your comments to enrich the scientific content of our work. In the new manuscript the only term used is vehicle.
- The reviewer strongly agrees that clean energy is needed, and ICE vehicles are replaced in this case. However, one important aspect to promote clean vehicles is because of the drop in cost and maturity of batteries. The authors are recommended to strengthen this argument in the introduction and include this reference to support this claim: Lai, C.S. and McCulloch, M.D., 2017. Levelized cost of electricity for solar photovoltaic and electrical energy storage. Applied Energy, 190, pp.191-203.
Thank you, we appreciate your comments. It was included this argument in the introduction as showed below.
“In addition, in recent years interest has increased due to technological advances. Lithium-ion batteries have increased their capacity, life time, and safety while their cost has decreased [3].”
Moreover, reference Lai, C.S. and McCulloch, M.D., 2017. Levelized cost of electricity for solar photovoltaic and electrical energy storage. Applied Energy, 190, pp.191-203. was included where can be consulted an in-depth look at this topic.
- English is a major concern with several grammatical errors. The paper needs to be proofread by native English speakers or authors should seek for professional language editing service. Some errors include: “In [9] is presented the switched capacitor” should be corrected to “Reference [9] presents….”, “In this scheme are necessary one capacitor and two switches to equalize two adjacent cells.”, “It is switched a capacitor between two adjacent cells”
Thank you, we appreciate your comments. The above grammatical errors were corrected in the new version of the manuscript. Moreover, the new version of the manuscript was proofreaded by a native english speaker.
- The reviewer questions the novelty and meaningfulness of this work. The claimed contribution of this work is: “A novel topology of RC-based equalizer is proposed to overcome the surge current in the cells without compromising the complexity of the controller. In this way, the cells are protected and it is improved the health of the system”. Generally, a snubber circuit can be used to suppress the inrush or surge current. Others have looked into ways for achieving this as follows: Mizanur, R., Khan, S., Rahman, A., Hrairi, M., Ferdaus, M.M. and Shahid, Z., 2014, November. A Battery Charge Balancing system with reducing inrush high spike current for electric vehicle. In 2014 IEEE International Conference on Smart Instrumentation, Measurement and Applications (ICSIMA) (pp. 1-6). IEEE. Kim, H.S., Park, K.B., Seong, H.W., Moon, G.W. and Youn, M.J., 2011, May. Flyback battery equalizer with voltage doubler rectifier. In 8th International Conference on Power Electronics-ECCE Asia (pp. 291-295). IEEE.
Thank you very much for your feedback. Indeed, a snubber circuit is a cheap and straightforward alternative to reduce the high current transients. Nevertheless, with the proposed topology is achieved a very flexible equalizer circuit. The unique function of the snubber is to limit the current while the proposed topology can emulate the operating principle of a switched resistor-based equalizer, a switched capacitor-based equalizer and also have the possibility to limit the current if exist conditions of high dispersion in voltage across the battery pack. A high-level controller can regulate the system to perform its strengths in every condition. Moreover, the element count is similar since the addition of a snubber circuit typically consists of a diode, a resistor and a capacitor. On the other hand, to achieve the proposed topology is needed a resistor and two MOSFETs. It is considered that taking into account the flexibility of the circuit when compared to the snubber; it is worth the implementation of the proposed topology. This explanation is included in the Introduction section
“In the literature, transient current peaks have been limited with snubber circuits [17,18]. This solution is quite cheap and straightforward. However, the sole purpose of this approach is to limit the current. This paper presents a topology that can emulate the operating principle of a switched resistor-based equalizer, a switched capacitor-based equalizer and also can limit the current if it is compromised the safety of the battery. A high-level controller can use this flexibility to perform the strengths of the system in every condition. In Section 2 the proposed architecture is presented and its modes of operation are explained; in Section 3 the steps to design the converter are illustrated; in Section 4 the proposed converter is compared to the classic topologies of capacitor-based and resistor-based BECs and in Section 5 the simulation results are shown. Finally, the conclusions are presented where the main generalizations of the paper are given.”
- The authors need to strengthen the literature review and in particular, the contribution of the work.
Thank you, we appreciate your comments. The literature review in the introduction was improved by adding 6 new references in the segments highlighted below:
“In addition, in recent years interest has increased due to technological advances. Lithium-ion batteries have increased their capacity, life time, and safety while their cost has decreased [3].”
“Finally, a current challenge is the real-time control since it presents a large amount of switches and therefore, a large amount of possibilities to decide the battery configuration [8].”
“Furthermore, reference [11] proposes an analog shunting scheme.”
“These reasons can lead to large transient voltages and temporal high dispersion of voltage across the battery pack [7,15].”
“In the literature, transient current peaks have been limited with snubber circuits [17,18]. This solution is quite cheap and straightforward.”
Moreover, the simulations of the switched resistor-based and switched capacitor-based equalizers were included in the Section 5 to strengthen the contribution of the work.
- Section 4 claims “to help to understand the benefits and limitations of the proposed equalizer with the other equalizers present in the literature.” Apart from the explanation, I suggest including some quantitative studies (e.g. total power loss) to support the proposed technique with comparison to the types of Equalizer identified in Table 3.Numerical values should be added to abstract.
Thank you very much for your feedback. In the new version of the manuscript were included the power losses equations of other hardware-based topologies. Moreover, it was summarized this topic including that parameter in Table 3 as suggested.
“Otherwise, Equation 8 represents the losses in the resistor-based equalizer. Where I is the charge current and R is the resistor where the excess of energy is burned. It is appreciated that presents a low efficiency when compared to the RC-based mode since R is much greater than the term RSV1+Ron_SPDT1+RDS(on)SR+RESR(C). Furthermore, Equation 9 illustrates the losses in the classic capacitor-based equalizer. It is assumed that the switching power losses are negligible. It can be noted that it is similar to the RC-based mode, but is suppressed the term RDS(on)SR. Hence, when compared to the RC-based mode, it is more efficient. It can be approximated that is 30% more efficient, since the on-resistance on the MOSFETs, in general, is greater than the ESR of the capacitor and the internal resistance of the cells.”
Moreover, the threshold current of the work was included in the abstract.
“The work was validated with a threshold current of 4A. However, it is possible design the system for any threshold current.”
Reviewer 2 Report
The paper presents an active method for a battery equalization of the cell voltages. It is referrred to as the RC-based equalizer. It combines two phases: an RC-phase and a C-based phase. The RC-based diagram is very simple to implement, which is an advantage.
Some comments are:
- When presenting the classification in Figure 1, the authors should explain the ones presented in the paper are for a hardware-based battery equalizer but they are other options as presented in their reference [12]. The authors should clearly indicate why the hardware-based solutions are considered more appropriate for EVs than the others.
- In Section 1, the authors state “This topology is especially suitable for battery banks with high dispersion voltage between cells.” Please, add an explanation about when this happens for EV batteries.
- Equations 2 and 3 should include the value of the capacitance. I_c =C dv/dt.
- In Section 4, the authors estimate the losses of their method. This should be compared with other hardware-based equalization techniques. Just the equations do not provide enough information.
- Section 5 should include the simulations with other hardware-based techniques to illustrate the advantages of the proposed method.
- The text should be revised by a native English speaker. It contains some typos.
Author Response
Reviewer 2. The paper presents an active method for a battery equalization of the cell voltages. It is referrred to as the RC-based equalizer. It combines two phases: an RC-phase and a C-based phase. The RC-based diagram is very simple to implement, which is an advantage.
Some comments are:
- When presenting the classification in Figure 1, the authors should explain the ones presented in the paper are for a hardware-based battery equalizer but they are other options as presented in their reference [12]. The authors should clearly indicate why the hardware-based solutions are considered more appropriate for EVs than the others.
We appreciate your comments to enrich the scientific content of our work. The proper explanation is presented in the introduction
“This classification corresponds to fixed battery architectures. In recent years a hot topic of research is the reconfigurable battery or software-defined battery. In this concept, the interconnections of the cells in a battery pack are reconfigured using switches. In this way, is achieved flexibility in the battery where its parameters such as the available power, operating voltage, etc. can be modified in real-time. This flexibility is used to accommodate the battery and load requirements. Nevertheless, since EVs require a large-scale battery pack, this technology presents more significant challenges. The weight associated with the switches and drivers can reach the 8% of the total system. Moreover, it is a limitation the presence of switches in the path of the power current that leads to the need for thermal management and negatively impacts the efficiency of the system. Furthermore, it is challenging the operation in real-time since the operational current is large. Finally, a current challenge is real-time control since it presents a large number of switches and, therefore, a large number of possibilities to decide the battery configuration [8].”
Furthermore, the reference 8 was included where the reader can consult more information about this topic
Muhammad, S.; Rafique, M.U.; Li, S.; Shao, Z.; Wang, Q.; Liu, X. Reconfigurable battery systems: a survey on hardware architecture and research challenges.ACM Trans. on Design Automation of Electronic Systems (TODAES) 2019,24, 1–27.
- In Section 1, the authors state “This topology is especially suitable for battery banks with high-dispersion voltage between cells.” Please, add an explanation about when this happens for EV batteries.
Thank you, we appreciate your comments. The explanation of the reasons that can lead to a high dispersion in voltage across the battery pack was rewritten as showed below.
“This topology is especially suitable for battery banks with high-dispersion voltage between cells. As stated in [12] the operating voltage is affected by internal parameters such as capacity and internal resistance. Otherwise, temperature and discharge/charge current are external parameters that affect the voltage. Therefore, this equalization variable does not reflect accurately the internal state of the cell [13]. Moreover, the voltage is affected if cells with a large difference in State of Health (SoH) or aging are used. These reasons can lead to large transient voltages and temporal high dispersion of voltage across the battery pack [7,14].”
In this way, the ideas are clearer for the reader. Moreover, the reference presented below was included where can be consulted another causes for voltage inconsistencies.
Feng, F.; Hu, X.; Hu, L.; Hu, F.; Li, Y.; Zhang, L. Propagation mechanisms and diagnosis of parameter inconsistency within Li-Ion battery packs. Renewable and Sustainable Energy Reviews, 2019,112, 102–113.
- Equations 2 and 3 should include the value of the capacitance. I_c =C dv/dt.
Thank you, we appreciate your comments. It was a typo in Equation 2. The new version of the manuscript includes the value of capacitance C in the equation. In Equation 3 this term is suppressed when substituting Equation 1 in 2. Moreover, R_eq was included in the denominator of Equation 3, since it was missing the term.
- In Section 4, the authors estimate the losses of their method. This should be compared with other hardware-based equalization techniques. Just the equations do not provide enough information.
Thank you very much for your feedback. In the new version of the manuscript were included the power losses equations of other hardware-based topologies. Moreover, it was summarized this topic including that parameter in Table 3.
“Otherwise, Equation 8 represents the losses in the resistor-based equalizer. Where I is the charge current and R is the resistor where the excess of energy is burned. It is appreciated that presents a low efficiency when compared to the RC-based mode since R is much greater than the term RSV1+Ron_SPDT1+RDS(on)SR+RESR(C). Furthermore, Equation 9 illustrates the losses in the classic capacitor-based equalizer. It is assumed that the switching power losses are negligible. It can be noted that it is similar to the RC-based mode, but is suppressed the term RDS(on)SR. Hence, when compared to the RC-based mode, it is more efficient. It can be approximated that is 30% more efficient, since the on-resistance on the MOSFETs, in general, is greater than the ESR of the capacitor and the internal resistance of the cells.”
- Section 5 should include the simulations with other hardware-based techniques to illustrate the advantages of the proposed method.
Thank you, we appreciate your comments. The simulations of the switched resistor-based and switched capacitor-based equalizers were included in the Section 5 as suggested.
- The text should be revised by a native English speaker. It contains some typos.
Thank you, we appreciate your comments. Moreover, the new version of the manuscript was proofreaded by a native english speaker.
Round 2
Reviewer 1 Report
The authors have revised the paper. I have the following comments.
Grammar still needs improvement. Change “A threshold current of 4A serves to validate the proposal achieving an increase of 35% in the equalization time.” to “the proposed method increases the equalization time by 35% for a threshold current of 4A”
Figures 17-19 should not be in conclusion.
Author Response
Reviewer 1. The authors have revised the paper. I have the following comments.
Grammar still needs improvement. Change “A threshold current of 4A serves to validate the proposal achieving an increase of 35% in the equalization time.” to “the proposed method increases the equalization time by 35% for a threshold current of 4A”
We appreciate your comments to enrich the scientific content of our work. In the new manuscript the proper change was done. Moreover, the paper was checked by the professional English editing services of MDPI (english-18221).
Figures 17-19 should not be in conclusion.
Thank you very much for your feedback. In the new manuscript, the figures are properly placed in Section 5.
Reviewer 2 Report
The authors have addressed most of my comments. However, there are still some changes to be applied:
- In Section 4, the authors should compare the power losses for a typical configuration. As it is now, there are only equations but the reader Will not know when (for which currents, for instance) the proposed equalization method is more convenient.
- The text still contains multiple typos and grammar errors.
Author Response
Reviewer 2. The authors have addressed most of my comments. However, there are still some changes to be applied:
In Section 4, the authors should compare the power losses for a typical configuration. As it is now, there are only equations but the reader Will not know when (for which currents, for instance) the proposed equalization method is more convenient.
We appreciate your comments to enrich the scientific content of our work. In Section 4 was included a typical configuration and is used this parameter to justify the selection of the proposed equalizer.
“A typical configuration is analyzed to put these equations in perspective. A common cell in this type of application is the 18650 battery, which has a maximum allowed current of 4A. Therefore, a safe equalization current is 3.5A. Furthermore, the conduction resistance of the elements RSV1,Ron_SPDT1,RESR(C) can be estimated at 110mΩ and R is established, in the previous section, to 3Ω. For these values, the power losses for the resistor-based equalizer is 36.75W. While in the case of the capacitor-based equalizer, a maximum instantaneous power losses of 1.3475W is obtained. In this case, since the current is a decreasing exponential function, the power losses will also behave in this way.
On the other hand, in the proposed equalizer, there is a first stage where the converter operates as an RC circuit. If a voltage difference between cells of 0.5V is present in the circuit, a maximum current peak of 0.16A is demanded from the cells as seen in Equation 3. In this way, the instantaneous power losses have a maximum peak of 2.84mW. When the voltage difference between the cells falls below a threshold, which guarantees a safe induced current in the cells, then the capacitor-based mode of operation of the proposed equalizer is activated. In this mode, a power loss equal to that of the classical capacitor-based scheme is obtained. It can be concluded that the proposed equalizer also limits the power losses in the circuit. Hence, the proposed topology should be used when there is a high voltage dispersion between cells.”
The text still contains multiple typos and grammar errors.
Thank you, we appreciate your comments. The paper was checked by the professional English editing services of MDPI (english-18221).
Round 3
Reviewer 2 Report
All my comments have been addressed.